# Dynamic Regret of Policy Optimization in Non-Stationary Environments

**Yingjie Fei**[1]   **Zhuoran Yang**[2]   **Zhaoran Wang**[1]   **Qiaomin Xie**[3]
[1] Northwestern University; yf275@cornell.edu, zhaoranwang@gmail.com
[2] Princeton University; zy6@princeton.edu
[3] Cornell University; qiaomin.xie@cornell.edu

## Abstract

We consider reinforcement learning (RL) in episodic MDPs with adversarial full-information reward feedback and unknown fixed transition kernels. We propose two model-free policy optimization algorithms, POWER and POWER++, and establish guarantees for their dynamic regret. Compared with the classical notion of static regret, dynamic regret is a stronger notion as it explicitly accounts for the non-stationarity of environments. The dynamic regret attained by the proposed algorithms interpolates between different regimes of non-stationarity, and moreover satisfies a notion of adaptive (near-)optimality, in the sense that it matches the (near-)optimal static regret under slow-changing environments. The dynamic regret bound features two components, one arising from exploration, which deals with the uncertainty of transition kernels, and the other arising from adaptation, which deals with non-stationary environments. Specifically, we show that POWER++ improves over POWER on the second component of the dynamic regret by actively adapting to non-stationarity through prediction. To the best of our knowledge, our work is the first dynamic regret analysis of model-free RL algorithms in non-stationary environments.

## 1   Introduction

Classical reinforcement learning (RL) literature often evaluates an algorithm by comparing its performance with that of the best *fixed* (i.e., stationary) policy in hindsight, where the difference is commonly known as regret. Such evaluation metric implicitly assumes that the environment is static so that it is appropriate to compare an algorithm to a single best policy. However, as we advance towards modern and practical RL problems, we face challenges arising in dynamic and non-stationary environments for which comparing against a single policy is no longer sufficient.

Two of the most prominent examples of RL for non-stationary environments are continual RL [30] and meta RL [16, 51] (and more broadly meta learning [20, 21]), which are central topics in the study of generalizability of RL algorithms. In these settings, an agent encounters a stream of tasks throughout time and aims to solve each task with knowledge accrued via solving previous tasks. The tasks can be very different in nature from each other, with potentially increasing difficulties. In particular, the reward mechanism may vary across tasks, and therefore requires the agent to adapt to the change of tasks. Another example of RL under non-stationary environments is human-machine interaction [23, 41]. This line of research studies how humans and machines (or robots) should interact or collaborate to accomplish certain goals. In one scenario, a human teaches a robot to complete a task by assigning appropriate rewards to the robot but without intervening its dynamics. The rewards from the human can depend on the stage of the learning process and the rate of improvement in the robot's behaviors. Therefore, the robot has to adjust its policy over time to maximize the rewards it receives.

In the above examples, it is uninformative to compare an algorithm with a fixed stationary policy, which itself may not perform well given the rapidly changing nature of environments. It is also unclear whether existing algorithms, designed for static environments and evaluated by the standard notion of regret, are sufficient for tackling non-stationary problems.

We aim to address these challenges in this paper. We consider the setting of episodic Markov decision processes (MDPs) with adversarial full-information reward feedback and unknown fixed transition kernels. We are interested in the notion of *dynamic* regret, the performance difference between an algorithm and the set of policies optimal for *individual* episodes in hindsight. For non-stationary RL, dynamic regret is a significantly stronger and more appropriate notion of performance measure than the standard (static) regret, but on the other hand more challenging for algorithm design and analysis. We propose two efficient, model-free policy optimization algorithms, POWER and POWER++. Under a mild regularity condition of MDPs, we provide dynamic regret analysis for both algorithms and we show that the regret bounds interpolate bewteen different regimes of non-stationarity. In particular, the bounds are of order $\tilde{O}(T^{1/2})$ when the underlying model is nearly stationary, matching with existing near-optimal static regret bounds. In that sense, our algorithms are *adaptively* near-optimal in slow-varying environments. To the best of our knowledge, we provide the first dynamic regret analysis for model-free RL algorithms under non-stationary environments.

Our dynamic regret bounds naturally decompose into two terms, one due to maintaining optimism and encouraging exploration in the face of uncertainty associated with the transition kernel, and the other due to the changing nature of reward functions. This decomposition highlights the two main components an RL algorithm needs in order to perform well in non-stationary environments: effective exploration under uncertainty and self-stabilization under drifting reward signals. Our second algorithm, POWER++, takes advantage of active prediction and improves over POWER in terms of the second term in the dynamic regret bounds.

**Our contributions.** The contributions of our work can be summarized as follows:

- We propose two model-free policy optimization algorithms, POWER and POWER++, for non-stationary RL with adversarial rewards;

- We provide dynamic regret analysis for both algorithms, and the regret bounds are applicable across all regimes of non-stationarity of the underlying model;

- When the environment is nearly stationary, our dynamic regret bounds are of order $\tilde{O}(T^{1/2})$ and match the near-optimal static regret bounds, thereby demonstrating the adaptive near-optimality of our algorithms in slow-changing environments.

**Related work.** Dynamic regret has been considered for RL in several papers. The work of [27] considers the setting of online MDP in which the transition kernel and reward function are allowed to change $l$ times, and the regret compares the algorithm against optimal policies for each of the $l + 1$ periods. It proposes UCRL2 with restart, which achieves an $\tilde{O}((l + 1)^{1/3}T^{2/3})$ regret where $T$ is the number of timesteps. The work of [22] considers the same setting and shows that UCRL2 with sliding windows achieves the same regret. Generalizing the previous settings, the work of [39] studies the setting where the changes of model is allowed to take place in every timestep. It proves that UCRL with restart achieves a regret of $\tilde{O}((B_r + B_p)^{1/3}T^{2/3})$ for sufficiently large $B_r, B_p > 0$, where $B_r$ and $B_p$ are the variations of rewards and transition kernels over the $T$ timesteps, respectively. The work of [13] proposes the sliding-window UCRL2 with confidence widening, which achieves an $\tilde{O}((B_r + B_p + 1)^{1/4}T^{3/4})$ regret; under additional regularity conditions, the regret can be improved to $\tilde{O}((B_r + B_p + 1)^{1/3}T^{2/3})$. A Bandit-over-RL algorithm is also provided by [13] to adaptively tune the UCRL2-based algorithm to achieve an $\tilde{O}((B_r + B_p + 1)^{1/4}T^{3/4})$ regret without knowing $B_r$ or $B_p$. The work [34] considers the setting of episodic MDPs in which reward functions and transition kernels get corrupted by an adversary in $K_0$ episodes. It proposes an algorithm called CRANE-RL that achieves a regret of $\tilde{O}(K_0\sqrt{T} + K_0^2)$. We remark that all the work discussed so far study model-based algorithms, and we refer interested readers to [40] for an excellent survey on the topic of RL in non-stationary environments. Dynamic regret has also been studied under the settings of multi-armed bandits [3, 6, 8, 11, 12, 31–33, 48], online convex optimization [7, 24–26, 44, 47, 49, 52, 55–62] and games [17]. Interestingly, the notion of dynamic regret is related to the exploitability of strategies in two-player zero-sum games [14]. We would also like to mention a series of papers that consider the

setting of non-stationary MDPs [1, 2, 10, 15, 19, 28, 36–38, 41, 45, 46, 53, 54], although they focus on static regret analysis.

**Notations.** For a positive integer $n$, we let $[n] := \{1, 2, \ldots, n\}$. We write $x^+ = \max\{x, 0\}$ for a scalar or vector $x$, where the maximum operator is applied elementwise. For two non-negative sequences $\{a_i\}$ and $\{b_i\}$, we write $a_i \lesssim b_i$ if there exists a universal constant $C > 0$ such that $a_i \leq C b_i$ for all $i$. We write $a_i \asymp b_i$ if $a_i \lesssim b_i$ and $b_i \lesssim a_i$. We use $\tilde{O}(\cdot)$ to denote $O(\cdot)$ while hiding logarithmic factors. We use $\|\cdot\|$ or $\|\cdot\|_2$ to denote the $\ell_2$ norm of a vector or spectral norm of a matrix, and $\|\cdot\|_1$ for the $\ell_1$ norm of a vector. We denote by $\Delta(\mathcal{X})$ the set of probability distributions supported on a discrete set $\mathcal{X}$. We define

$$\Delta(\mathcal{X} \mid \mathcal{Y}, H) := \left\{ \{\pi_h(\cdot \mid \cdot)\}_{h \in [H]} : \pi_h(\cdot \mid y) \in \Delta(\mathcal{X}) \text{ for any } y \in \mathcal{Y} \text{ and } h \in [H] \right\}$$

for any set $\mathcal{Y}$ and horizon length $H \in \mathbb{Z}_{>0}$. For $p_1, p_2 \in \Delta(\mathcal{X})$, we define $D_{\mathrm{KL}}(p_1 \| p_2)$ to be the KL divergence between $p_1$ and $p_2$, that is, $D_{\mathrm{KL}}(p_1 \| p_2) := \sum_{x \in \mathcal{X}} p_1(x) \log \left( \frac{p_1(x)}{p_2(x)} \right)$.

## 2 Preliminaries

### 2.1 Episodic MDPs and dynamic regret

In this paper, we study RL in non-stationary environments via episodic MDPs with adversarial full-information reward feedback and unknown fixed transition kernels. An episodic MDP is defined by the state space $\mathcal{S}$, the action space $\mathcal{A}$, the length $H$ of each episode, the transition kernels $\{\mathcal{P}_h(\cdot \mid \cdot, \cdot)\}_{h \in [H]}$ and the reward functions $\{r_h^k : \mathcal{S} \times \mathcal{A} \to [0, 1]\}_{(k,h) \in [K] \times [H]}$. We assume that the reward functions are deterministic and potentially different across episodes, and that both $\mathcal{S}$ and $\mathcal{A}$ are discrete sets of sizes $S := |\mathcal{S}|$ and $A := |\mathcal{A}|$, respectively.

An agent interacts with the MDP through $K$ episodes without knowledge of $\{\mathcal{P}_h\}$. At the beginning of episode $k$, the environment provides an arbitrary state $s_1^k$ to the agent and chooses reward functions $\{r_h^k\}_{h \in [H]}$. The choice of the reward functions is possibly adversarial and may depend on the history of the past $(k-1)$ episodes. In step $h$ of episode $k$, the agent observes state $s_h^k$ and then takes an action $a_h^k$, upon which the environment transitions to the next state $s_{h+1}^k \sim \mathcal{P}(\cdot \mid s_h^k, a_h^k)$. At the same time, the environment also reveals the reward function $r_h^k$ to the agent, and the agent receives the reward $r_h^k(s_h^k, a_h^k)$ (known as the full-information setting). At step $H + 1$, the agent observes state $s_{H+1}^k$ but does not take any action (therefore receiving no reward), and episode $k$ is completed. We denote by $T := KH$ the total number of steps taken throughout the $K$ episodes.

For any fixed policy $\pi = \{\pi_h\}_{h \in [H]} \in \Delta(\mathcal{A} \mid \mathcal{S}, H)$ and any $(k, h, s, a) \in [K] \times [H] \times \mathcal{S} \times \mathcal{A}$, we define the value function $V_h^{\pi,k} : \mathcal{S} \to \mathbb{R}$ as

$$V_h^{\pi,k}(s) := \mathbb{E}_\pi \left[ \sum_{i=h}^H r_i^k(s_i, a_i) \,\middle|\, s_h = s \right],$$

and the corresponding action-value function $Q_h^{\pi,k} : \mathcal{S} \times \mathcal{A} \to \mathbb{R}$ as

$$Q_h^{\pi,k}(s, a) := \mathbb{E}_\pi \left[ \sum_{i=h}^H r_i^k(s_i, a_i) \,\middle|\, s_h = s, a_h = a \right].$$

Here, the expectation $\mathbb{E}_\pi[\cdot]$ is taken over the randomness of the state-action tuples $\{(s_h, a_h, s_{h+1})\}_{h \in [H]}$, where the action $a_h$ is sampled from the policy $\pi_h(\cdot \mid s_h)$ and the next state $s_{h+1}$ is sampled from the transition kernel $\mathcal{P}_h(\cdot \mid s_h, a_h)$. The Bellman equation is given by

$$Q_h^{\pi,k}(s, a) = r_h^k + \mathbb{P}_h V_{h+1}^{\pi,k}, \qquad V_h^{\pi,k}(s) := \left\langle Q_h^{\pi,k}, \pi_h \right\rangle_{\mathcal{A}}, \qquad V_{H+1}^\pi(s) = 0. \tag{1}$$

In Equation (1), we use $\langle \cdot, \cdot \rangle_{\mathcal{A}}$ to denote the inner product over $\mathcal{A}$ and we will omit the subscript $\mathcal{A}$ in the sequel when appropriate; we also define the operator

$$(\mathbb{P}_h f)(s, a) := \mathbb{E}_{s' \sim \mathcal{P}_h(\cdot \mid s, a)}[f(s')]$$

for any function $f : \mathcal{S} \to \mathbb{R}$.

Under the setting of episodic MDPs, the agent aims to approximate the optimal non-stationary policy by interacting with the environment. Let $\pi^{*,k} = \mathrm{argmax}_{\pi \in \Delta(\mathcal{A}|\mathcal{S},H)} V_1^{\pi,k}(s_1^k)$ be the optimal policy of episode $k$, and suppose that the agent executes policy $\pi^k$ in episode $k$. The difference in values between $V_1^{\pi^k,k}(s_1^k)$ and $V_1^{\pi^{*,k},k}(s_1^k)$ serves as the regret or the sub-optimality of the agent's policy $\pi^k$ in episode $k$. Therefore, the *dynamic regret* for $K$ episodes is defined as

$$\text{D-Regret}(K) := \sum_{k \in [K]} \left[ V_1^{\pi^{*,k},k}(s_1^k) - V_1^{\pi^k,k}(s_1^k) \right]. \tag{2}$$

Dynamic regret is a stronger notion than the classical regret measure found in the literature of online learning and reinforcement learning, which is also known as static regret and defined as

$$\text{Regret}(K) := \sum_{k \in [K]} \left[ V_1^{\pi^*,k}(s_1^k) - V_1^{\pi^k,k}(s_1^k) \right], \tag{3}$$

where $\pi^* = \mathrm{argmax}_{\pi \in \Delta(\mathcal{A}|\mathcal{S},H)} \sum_{k \in [K]} V_1^{\pi,k}(s_1^k)$. In words, dynamic regret compares the agent's policy to the optimal policy of *each individual* episode in the hindsight, while static regret compares the agent's policy to only the optimal fixed policy over all episodes combined. Therefore, the notion of dynamic regret is a more natural measure of performance under non-stationary environments. It is clear that dynamic regret always upper bounds static regret:

$$\text{D-Regret}(K) = \sum_{k \in [K]} \left[ \max_{\pi \in \Delta(\mathcal{A}|\mathcal{S},H)} V_1^{\pi,k}(s_1^k) - V_1^{\pi^k,k}(s_1^k) \right]$$

$$\geq \max_{\pi \in \Delta(\mathcal{A}|\mathcal{S},H)} \sum_{k \in [K]} \left[ V_1^{\pi,k}(s_1^k) - V_1^{\pi^k,k}(s_1^k) \right] = \text{Regret}(K).$$

When $\{\pi^{*,k}\}$ happen to be identical for all episodes $k \in [K]$, dynamic regret reduces to static regret.

## 2.2 Model assumptions

For any policy $\pi$, step $h \in [H]$ and states $s, s' \in \mathcal{S}$, we denote by $\mathcal{P}_h^\pi(s' \mid s)$ the probability of transitioning from $s$ to $s'$ in step $h$ when policy $\pi$ is executed, i.e., $\mathcal{P}_h^\pi(s' \mid s) := \sum_{a \in \mathcal{A}} \mathcal{P}_h(s' \mid s, a) \cdot \pi_h(a \mid s)$. The quantity $\mathcal{P}_h^\pi$ is also known as the visitation measure of $\pi$ at state $s$ and step $h$. For any pair of policies $\pi$ and $\pi'$, we define the shorthands

$$\|\pi_h - \pi_h'\|_\infty := \max_{s \in \mathcal{S}} \|\pi_h(\cdot \mid s) - \pi_h'(\cdot \mid s)\|_1,$$

$$\|\mathcal{P}_h^\pi - \mathcal{P}_h^{\pi'}\|_\infty := \max_{s \in \mathcal{S}} \|\mathcal{P}_h^\pi(\cdot \mid s) - \mathcal{P}_h^{\pi'}(\cdot \mid s)\|_1.$$

The following assumption stipulates that the visitation measures are smooth with respect to policies.

**Assumption 1** (Smooth visitation measures). *We assume that there exists a universal constant $C > 0$ such that $\|\mathcal{P}_h^\pi - \mathcal{P}_h^{\pi'}\|_\infty \leq C \cdot \|\pi_h - \pi_h'\|_\infty$ for all $h \in [H]$ and all pairs of policies $\pi, \pi'$.*

Assumption 1 states that the visitation measures do not change drastically when similar policies are executed. This notion of smoothness in visitation measures also appears in [41] in the context of two-player games.

*Remark* 1. Assumption 1 can in fact be relaxed to $\|\mathcal{P}_h^\pi - \mathcal{P}_h^{\pi'}\|_\infty \leq C \cdot \|\pi_h - \pi_h'\|_\infty$ for all $h \in [H]$ and $C = O(T^\alpha)$ that holds for all $\alpha > 0$ (i.e., the Lipschitz parameter $C$ is sub-polynomial in $T$), and our algorithms and results remain the same. We choose to instead require $C > 0$ to be a universal constant for clear exposition.

Next, we introduce several measures of changes in MDPs and algorithms. Define

$$P_T := \sum_{k \in [K]} \sum_{h \in [H]} \|\pi_h^{*,k} - \pi_h^{*,k-1}\|_\infty, \tag{4}$$

where we set $\pi_h^{*,0} = \pi_h^{*,1}$ for $h \in [H]$. Note that $P_T$ measures the total variation in the optimal policies of adjacent episodes. Oftentimes, algorithms are designed to estimate the optimal policies

$\{\pi^{*,k}\}_{k \in [K]}$ by estimating action-value functions $\{Q^{\pi^{*,k},k}\}_{k \in [K]}$ via iterates $\{Q^k\}_{k \in [K]}$. For such algorithms, we define

$$D_T := \sum_{k \in [K]} \sum_{h \in [H]} \max_{s \in \mathcal{S}} \|Q_h^k(s, \cdot) - Q_h^{k-1}(s, \cdot)\|_\infty^2, \tag{5}$$

where we set $Q_h^0 = Q_h^1$ for $h \in [H]$. Therefore, the quantity $D_T$ computes total variation in algorithmic iterates $\{Q^k\}$. The notions of $P_T$ and $D_T$ are also used in the work of [7, 24, 25, 43, 62] and are known as *variation budgets* or *path lengths*. We assume that we have access to quantities $P_T$ and $D_T$ or their upper bounds via an oracle, but we do not know $\{\pi^{*,k}\}$. Such assumptions are standard in non-stationary RL and online convex optimization [7, 22, 27, 39, 42, 43].

## 2.3 Connections with popular RL paradigms

We briefly discuss how the setting introduced in Section 2.1 is related to several popular paradigms of RL. In certain settings of continual and meta RL, an agent needs to solve tasks one after another in the same physical environment and receives rewards for each task commensurate to the agent's performance in solving the task. A task can therefore be seen as an episode in our episodic setting. Since the tasks are presented and solved within the same physical environment, it is sufficient to assume a fixed transition model as we do in Section 2.1. On the other hand, the tasks to be solved by the agent can be substantially different from each other in reward mechanism, as such detail of each task is potentially determined by the agent's performance in all previous tasks. This suggests that the rewards of the tasks are possibly non-stationary, corresponding to the quantities $\{r_h^k\}$ in our setting.

Our setting can also be viewed as a high-level abstraction for human-machine interaction. As in the example discussed in Section 1, a human guides a robot (the learner) to accomplish certain tasks by only presenting rewards according to the performance of the robot. Here, we can think of the period in between two presented rewards as an episode in our setting. We may also set the physical state of the robot as the state of our model, thus implying a fixed state transition from the robot's perspective. Moreover, the rewards are controlled by the human in a way that possibly depends on time and history of the robot's performance, which corresponds to our assumption on $\{r_h^k\}$.

# 3 Algorithms

In this section, we present two efficient and model-free algorithms: **P**olicy **O**ptimization **W**ith **PE**riodic **R**estart (POWER) and its enhanced version, POWER++. Let us introduce some additional notations before proceeding. We set $d = |\mathcal{S}||\mathcal{A}|$, and let $\phi(s, a)$ be the canonical basis of $\mathbb{R}^d$ corresponding to the state-action pair $(s, a) \in \mathcal{S} \times \mathcal{A}$: that is, the $(s', a')$-th entry of $\phi(s, a)$ equals to 1 if $(s, a) = (s', a')$ and 0 otherwise.

## 3.1 POWER

We present our first algorithm, POWER, in Algorithm 1. Algorithm 1 is inspired by the work of [9, 18]. It mainly consists of a policy update and a policy evaluation step. The policy update step in Line 7 is equivalent to solving the following optimization problem:

$$\pi^k = \operatorname*{argmax}_{\pi \in \Delta(\mathcal{A} \mid \mathcal{S}, H)} L_{k-1}(\pi) - \frac{1}{\alpha} \mathbb{E}_{\pi^{k-1}} \left[ \left. \sum_{h \in [H]} D_{\mathrm{KL}}(\pi_h(\cdot \mid s_h) \| \pi_h^{k-1}(\cdot \mid s_h)) \right| s_1 = s_1^k \right], \tag{6}$$

where

$$L_{k-1}(\pi) := V_1^{\pi^{k-1},k-1}(s_1^k)$$
$$+ \mathbb{E}_{\pi^{k-1}} \left[ \left. \sum_{h \in [H]} \left\langle Q_h^{\pi^{k-1},k-1}(s_h, \cdot), \pi_h(\cdot \mid s_h) - \pi_h^{k-1}(\cdot \mid s_h) \right\rangle \right| s_1 = s_1^k \right]$$

is a local linear approximation of $V_1^{\pi,k-1}(s_1^k)$ at $\pi = \pi^{k-1}$. In view of Equation (6), we observe that the policy update step can be seen as a mirror descent (MD) step with KL divergence as the Bregman

---

**Algorithm 1** POWER

---

**Input:** Confidence level $\delta$, number of episodes $K$, restart cycle length $\tau$, step size $\alpha$, regularization factor $\lambda$ and bonus multiplier $\beta$

1: **for** episode $k = 1, \ldots, K$ **do**
2:      Receive the initial state $s_1^k$
3:      **if** $k \mod \tau = 1$ **then**                                         ▷ periodic restart
4:          Set $\{Q_h^{k-1}\}_{h \in [H]}$ as zero functions and $\{\pi_h^{k-1}\}_{h \in [H]}$ as uniform distributions on $\mathcal{A}$
5:      **end if**
6:      **for** step $h = 1, 2, \ldots, H$ **do**                                         ▷ policy update
7:          Update the policy by $\pi_h^k(\cdot \mid \cdot) \propto \pi_h^{k-1}(\cdot \mid \cdot) \cdot \exp\{\alpha \cdot Q_h^{k-1}(\cdot, \cdot)\}$
8:          Take action $a_h^k \sim \pi_h^k(\cdot \mid s_h^k)$
9:          Observe the reward function $r_h^k(\cdot, \cdot)$ and receive the next state $s_{h+1}^k$
10:     **end for**
11:     Compute $\{Q_h^k\}$ by EvaluatePolicy$(k, \{r_h^k\}, \{\pi_h^k\}, \lambda, \beta)$           ▷ policy evaluation
12: **end for**

---

divergence. The policy evaluation step in Line 11 estimates value functions of each step. To that end, it invokes a subroutine, EvaluatePolicy, which computes the intermediate estimates $w_h^k$ as the solution of the following regularized least-squares problem

$$w_h^k \leftarrow \underset{w \in \mathbb{R}^d}{\operatorname{argmin}} \sum_{t \in [k-1]} (V_{h+1}^k(s_{h+1}^t) - \phi(s_h^t, a_h^t)^\top w)^2 + \lambda \cdot \|w\|_2^2.$$

This step can be efficiently computed by taking the sample mean of $\{V_{h+1}^k(s_{h+1}^t)\}_{t \in [k-1]}$. In fact, one has

$$w_h^k(s, a) = \phi(s, a)^\top w_h^k = \sum_{s' \in \mathcal{S}} \frac{N_h^k(s, a, s')}{N_h^k(s, a) + \lambda} \cdot V_{h+1}^k(s'),$$

for each $(s, a)$, where the function $N_h^k$ counts the number of times each tuple $(s, a, s')$ or $(s, a)$ has been visited by the algorithm at step $h$ prior to episode $k$. To facilitate exploration in the face of uncertainties, EvaluatePolicy additionally defines a bonus term $\Gamma_h^k(s, a) \propto [N_h^k(s, a)]^{-1/2}$ for each state-action pair $(s, a)$. The estimated action-value function is then set as $Q_h^k = r_h^k + w_h^k + \Gamma_h^k$. We provide the detailed implementation of the subroutine EvaluatePolicy in Algorithm 3 in Appendices.

In addition to updating and evaluating policy, Algorithm 1 features a periodic restart mechanism, which resets its policy estimate every $\tau$ episodes. Restart mechanisms have been used to handle non-stationarity in RL [27, 39] and related problems including bandits [6], online convex optimization [7, 26] and games [17, 41]. Intuitively, by employing the restart mechanism, Algorithm 1 is able to stabilize its iterates against non-stationary drift in the learning process due to adversarial reward functions. We remark that our Algorithm 1 is very different from those used in the existing non-stationary RL literature. Notably, Algorithm 1 is model-free, which is more efficient than the model-based algorithms proposed in e.g., [12, 22, 27, 34, 39], with respect to both time and space complexities.

## 3.2 POWER++

Instead of only passively tackling non-stationarity, we may enhance our algorithms with active prediction of the environment. Optimistic mirror descent (OMD) provides exactly such prediction functionality via the so-called predictable sequences. It is well-known in the online learning literature that OMD provides improved regret guarantees than MD algorithm [42, 43]. First proposed by [35] under the name "mirror-prox", OMD maintains a sequence of main and intermediate iterates. Through the predictable sequences in intermediate iterates, it exploits certain structures of the problem at hand, and therefore achieve better theoretical guarantees. We incorporate predictable sequences into POWER and arrive at an enhanced algorithm, POWER++, which is presented in Algorithm 2.

In Algorithm 2, Lines 8 and 12 together form the OMD steps. Line 10 estimates the intermediate action-value function $Q_h^{k-1/2}$ to be used in the second OMD step (Line 12). The series of iterates $\{Q_h^{k-1}\}$ in Line 8 is the so-called predictable sequence in OMD. Note that we do not execute the

---

**Algorithm 2** POWER++

---

**Input:** Confidence level $\delta$, number of episodes $K$, restart cycle length $\tau$, step size $\alpha$, regularization factor $\lambda$ and bonus multiplier $\beta$

1: Set $\{r_h^0\}_{h \in [H]}$ as zero functions
2: **for** episode $k = 1, \dots, K$ **do**
3:      Receive the initial state $s_1^k$
4:      **if** $k \mod \tau = 1$ **then**                                  $\triangleright$ periodic restart
5:          Set $\{Q_h^{k-1}\}_{h \in [H]}$ as zero functions and $\{\pi_h^{k-1}\}_{h \in [H]}$ as uniform distributions on $\mathcal{A}$
6:      **end if**
7:      **for** step $h = 1, 2, \dots, H$ **do**                           $\triangleright$ intermediate policy update
8:          Update the policy by $\pi_h^{k-1/2}(\cdot \mid \cdot) \propto \pi_h^{k-1}(\cdot \mid \cdot) \cdot \exp\{\alpha \cdot Q_h^{k-1}(\cdot, \cdot)\}$
9:      **end for**
10:      Compute $\{Q_h^{k-1/2}\}$ by EvaluatePolicy$(k, \{r_h^{k-1}\}, \{\pi_h^{k-1/2}\}, \lambda, \beta)$
                                                    $\triangleright$ intermediate policy evaluation
11:      **for** step $h = 1, 2, \dots, H$ **do**                               $\triangleright$ main policy update
12:          Update the policy by $\pi_h^k(\cdot \mid \cdot) \propto \pi_h^{k-1}(\cdot \mid \cdot) \cdot \exp\{\alpha \cdot Q_h^{k-1/2}(\cdot, \cdot)\}$
13:          Take action $a_h^k \sim \pi_h^k(\cdot \mid s_h^k)$
14:          Observe the reward function $r_h^k(\cdot, \cdot)$ and receive the next state $s_{h+1}^k$
15:      **end for**
16:      Compute $\{Q_h^k\}$ by EvaluatePolicy$(k, \{r_h^k\}, \{\pi_h^k\}, \lambda, \beta)$             $\triangleright$ main policy evaluation
17: **end for**

---

intermediate policy $\pi^{k-1/2}$ in the first (and intermediate) OMD step (Line 8), which is only used to compute the intermediate value estimates $\{V_h^{k-1/2}\}$. Rather, we execute the policy $\pi^k$ updated by the second (and main) OMD step. Finally, we remark that both Algorithms 1 and 2 have polynomial space and time complexities in $S$, $A$ and $T$.

## 4   Main results

To help with the presentation of our main results, we define the thresholding operator $\Pi_{[a,b]}(x) := \max\{\min\{x, b\}, a\}$ and we adopt the convention that $x/0 = \infty$ for $x \in \mathbb{R}$. We also define $L := \left\lceil \frac{K}{\tau} \right\rceil$ to be the number of restarts that take place in Algorithm 1 or 2. The following theorem gives an upper bound for the dynamic regret incurred by Algorithm 1.

**Theorem 1** (Upper bound for Algorithm 1). *Under Assumption 1, for any $\delta \in (0, 1]$, with probability at least $1 - \delta$ and the choice of $\lambda = 1$, $\alpha = \sqrt{\frac{L \log A}{K H^2}}$, $\tau = \Pi_{[1, K]} \left( \left\lfloor \left( \frac{T \sqrt{\log A}}{H P_T} \right)^{2/3} \right\rfloor \right)$ and $\beta = C_\beta H \sqrt{S \log(dT/\delta)}$ (for some universal constant $C_\beta > 0$) in Algorithm 1, the dynamic regret of Algorithm 1 is bounded by*

$$\text{D-Regret}(K) \lesssim \sqrt{H^3 S^2 A T \cdot \log^2(dT/\delta)} + \begin{cases} \sqrt{H^3 T \log A}, & \text{if } 0 \le P_T \le \sqrt{\frac{\log A}{K}}, \\ \left( H^2 T \sqrt{\log A} \right)^{2/3} P_T^{1/3}, & \text{if } \sqrt{\frac{\log A}{K}} \le P_T \lesssim K \sqrt{\log A}, \\ H^2 P_T, & \text{if } P_T \gtrsim K \sqrt{\log A}. \end{cases}$$

*The result also holds if we replace $P_T$ in the above with its upper bound. When the upper bounds on* D-Regret$(K)$ *exceed $T$, we have* D-Regret$(K) \le T$.

The proof is given in Appendix C. The regret bound in Theorem 1 interpolates smoothly throughout three regimes of $P_T$:

- Small $P_T$: when $0 \le P_T \le \sqrt{\frac{\log A}{K}}$, the dynamic regret scales as $\tilde{O}(T^{1/2})$ and subsumes the static regret results in [9, 18] under the full-information setting. In view of [4], this bound is also nearly optimal (up to polynomial factors of $H$, $S$ and $A$). Therefore, our bound in Theorem 1 is *adaptively* near-optimal under small $P_T$;

- Moderate $P_T$: when $\sqrt{\frac{\log A}{K}} \leq P_T \lesssim K\sqrt{\log A}$, we obtain a dynamic regret of order $\tilde{O}(T^{2/3}P_T^{1/3})$, which is $\tilde{O}(T^{2/3})$ if $P_T = O(1)$ and sub-linear in $T$ if $P_T = o(K)$. Similar $\tilde{O}(T^{2/3})$ bounds have been achieved by model-based algorithms in [13, 22, 27, 39], which are less efficient than our model-free algorithms in both time and space complexities;

- Large $P_T$: when $P_T \gtrsim K\sqrt{\log A}$, the model is highly non-stationary and Algorithm 1 incurs a linear regret in $T$.

In addition, the dynamic regret bound in Theorem 1 can be seen as a combination of two parts. The first is the cost paid for being optimistic and due to sum of bonus terms $\{\Gamma_h^k\}$ in Algorithm 3 (see Equation (15) in the proof for details). This part is necessary to enforce optimism in the face of uncertainty generated by the transition kernels and is key to effective exploration. The second part is the error caused by non-stationarity of reward functions and depends on $P_T$. Such decomposition is not available in the dynamic regret analysis of online convex optimization problems where MD/OMD-based algorithms have been widely applied. In particular, the dynamic regret bound for online optimization lacks the term due to bonus as it does not require exploration, which is nevertheless a key component underlying RL algorithms that provably explore.

Next we present a result for Algorithm 2.

**Theorem 2** (Upper bound for Algorithm 2). *Under Assumption 1, for any $\delta \in (0,1]$, with probability at least $1 - \delta$ and the choice of $\lambda = 1$, $\alpha = \sqrt{\frac{LH\log A}{D_T}}$, $\tau = \Pi_{[1,K]}\left(\left\lfloor\left(\frac{\sqrt{D_T \cdot T \log A}}{H^2 P_T}\right)^{2/3}\right\rfloor\right)$ and $\beta = C_\beta H\sqrt{S\log(dT/\delta)}$ (for some universal constant $C_\beta > 0$) in Algorithm 2, the dynamic regret of Algorithm 2 is bounded by*

$$\text{D-Regret}(K) \lesssim \sqrt{H^3 S^2 AT \cdot \log^2(dT/\delta)} + \begin{cases} \sqrt{D_T \cdot H\log A}, & \text{if } 0 \leq P_T \leq \sqrt{\frac{D_T \cdot \log A}{K^2 H^3}}, \\ \left(H\sqrt{D_T \cdot T\log A}\right)^{2/3}P_T^{1/3}, & \text{if } \sqrt{\frac{D_T \cdot \log A}{K^2 H^3}} \leq P_T \lesssim \frac{\sqrt{D_T \cdot T\log A}}{H^2}, \\ H^2 P_T, & \text{if } P_T \gtrsim \frac{\sqrt{D_T \cdot T\log A}}{H^2}. \end{cases}$$

*The result also holds if we replace $P_T$ and $D_T$ in the above with their upper bounds. When the upper bounds on $\text{D-Regret}(K)$ exceed $T$, we have $\text{D-Regret}(K) \leq T$.*

The proof is given in Appendix D. A few remarks about Theorem 2 are in order. Similar to Theorem 1, the result in Theorem 2 interpolates across three regimes depending on the magnitude of $P_T$, and decomposes into two terms respectively arising from the uncertainties of transition kernels and non-stationarity of reward functions. Moreover, thanks to the OMD steps in Algorithm 2 that actively make predictions via predictable sequence $\{Q_h^{k-1}\}$, the bound in Theorem 2 is strictly better than that in Theorem 1 in view of the fact that $D_T \lesssim KH^3$. When $P_T$ is moderate, i.e., $\sqrt{\frac{D_T \cdot \log A}{K^2 H^3}} \leq P_T \lesssim \frac{\sqrt{D_T \cdot T\log A}}{H^2}$, the dynamic regret bound in Theorem 2 is of order $\tilde{O}(T^{1/3}D_T^{1/3}P_T^{1/3})$, which is similar to the result of [26, Theorem 3] obtained for online optimization problems. Regret bounds that depend on $D_T$, the variation of predictable sequences, have also appeared in [42, 43], although for static regret and online optimization problems.

**Technical highlights.** A central step of our dynamic regret analysis is to control the expected performance difference between the estimated policies $\{\pi^k\}$ and the optimal $\{\pi^{*,k}\}$, defined as

$$\sum_{k\in[K]}\sum_{h\in[H]} \mathbb{E}_{\pi^{*,k}}\left[\left\langle Q_h^k(s_h,\cdot), \pi_h^{*,k}(\cdot\mid s_h) - \pi_h^k(\cdot\mid s_h)\right\rangle \;\middle|\; s_1 = s_1^k\right].$$

Note the the expectation is taken over $\{\pi^{*,k}\}$ which may vary over episodes $k$. For static regret, i.e., when $\pi^{*,k} \equiv \pi^*$ for $k \in [K]$, we may control the above term by a standard telescoping argument, which is not viable for dynamic regret analysis. Instead, we decompose the above expectation into $\mathbb{E}_{\pi^{*,k}}[\cdot] = \mathbb{E}_{\pi^{*,k_0}}[\cdot] + \mathbb{E}_{\pi^{*,k} - \pi^{*,k_0}}[\cdot]$. Here, $k_0 < k$ is the episode in which restart takes place most recently prior to episode $k$. The first expectation $\mathbb{E}_{\pi^{*,k_0}}[\cdot]$ is taken over $\pi^{*,k_0}$, which stays constant for the period from $k_0$ to the next restart. Therefore, we may apply a customized telescoping argument to each period between restarts. The second expectation $\mathbb{E}_{\pi^{*,k} - \pi^{*,k_0}}[\cdot]$ from the decomposition

involves the difference $\pi^{*,k} - \pi^{*,k_0}$ and can be bounded by $P_T$. See Lemmas 3 and 4 in Appendices, respectively, for details of controlling the two expectations. Furthermore, it is noteworthy that the restart cycle length $\tau$ plays an important role of balancing the tradeoffs that 1) the optimal policies between two adjacent restarts are relatively stationary among themselves so that the algorithm is compared to stable benchmarks, and that 2) there are not too many restarts so that the sub-optimality of algorithm do not grow too fast when combined over periods in between restarts.

**Comparison with existing results.** We compare the results in Theorems 1 and 2 to those in [13], which is so far state-of-the-art in dynamic regret analysis for non-stationary RL. First, our model-free algorithms are more efficient than the model-based algorithm in [13] that is adapted from UCRL2 and requires solving linear programs in each timestep. Second, our bounds in Theorems 1 and 2 are on the near-optimal order $\tilde{O}(T^{1/2})$ when $P_T$ is sufficiently small, whereas the results in [13] are of order $\tilde{O}(T^{2/3})$. On the other hand, [13] studies a more general setting where the transition kernel of the MDP is allowed to vary adversarially in each timestep. It also provides a procedure to adaptively tune its UCRL2-based algorithm to achieve an $\tilde{O}(T^{3/4})$ regret without knowledge of variations such as $P_T$.

## Broader Impact

This work provides novel algorithms and analysis for non-stationary RL, which is the foundation of several important RL paradigms including continual/meta RL and human-machine interaction. We present two efficient and model-free policy optimization algorithms for episodic MDPs with adversarial reward functions and fixed unknown transitions. For both algorithms, we provide dynamic regret bounds that interpolate between different regimes of non-stationarity of the underlying model. We show that our bounds achieve the near-optimal $\tilde{O}(T^{1/2})$ order and are adaptively near-optimal in slow-changing environments. To the best of our knowledge, our work provides the first dynamic regret analysis for model-free algorithms in non-stationary RL.

## Acknowledgments and Disclosure of Funding

Q. Xie is partially supported by NSF grant 1955997.

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
