[Supplementary Material]

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

# Appendices

## A    Implementation of EvaluatePolicy

---
**Algorithm 3** EvaluatePolicy

---
**Input:** Episode index $k$, reward functions $\{r_h\}$, policies $\{\pi_h\}$, regularization factor $\lambda$ and bonus multiplier $\beta$
**Output:** Updated Q-values $\{Q_h\}$
 1: Initialize $V_{H+1}$ as a zero function
 2: **for** step $h = H, H-1, \ldots, 1$ **do**
 3: $\quad$ $\Lambda_h \leftarrow \sum_{t \in [k-1]} \phi(s_h^t, a_h^t)\phi(s_h^t, a_h^t)^\top + \lambda \cdot \mathbf{I}$
 4: $\quad$ $w_h \leftarrow (\Lambda_h)^{-1} \sum_{t \in [k-1]} \phi(s_h^t, a_h^t) \cdot V_{h+1}(s_{h+1}^t)$
 5: $\quad$ $\Gamma_h(\cdot, \cdot) \leftarrow \beta \cdot [\phi(\cdot, \cdot)^\top (\Lambda_h)^{-1} \phi(\cdot, \cdot)]^{1/2}$
 6: $\quad$ $Q_h(\cdot, \cdot) \leftarrow r_h(\cdot, \cdot) + \min\{\phi(\cdot, \cdot)^\top w_h + \Gamma_h(\cdot, \cdot), H - h\}^+$
 7: $\quad$ $V_h(\cdot) \leftarrow \langle Q_h(\cdot, \cdot), \pi_h(\cdot \mid \cdot) \rangle_{\mathcal{A}}$
 8: **end for**

---

In Algorithm 3, the tuples $\{(s_h^t, a_h^t)\}_{t \in [k-1]}$ are state-action pairs visited by Algorithm 1 or 2 before episode $k$.

## B    Proofs of technical lemmas

Recall that $L := \lceil \frac{K}{\tau} \rceil$. Algorithm 1 divides $K$ episodes into $L$ periods, and at the the beginning of each period it resets its Q-value and policy estimates. Each period contains $\tau$ episodes, except for the last one, which consists of at most $\tau$ episodes. For ease of notations, we assume that the last period has exactly $\tau$ episodes. Our proof can be easily extended to the case where the last period has fewer than $\tau$ episodes.

### B.1    Regret decomposition

For any $(k, h, s) \in [K] \times [H] \times \mathcal{S}$, we define the model prediction error

$$\iota_h^k := r_h^k + \mathbb{P}_h V_{h+1}^k - Q_h^k. \tag{7}$$

We have the following decomposition of the dynamic regret (2).

**Lemma 1.** We have

$$\text{D-Regret}(K) = \sum_{l \in [L]} \sum_{k=(l-1)\tau+1}^{l\tau} \sum_{h \in [H]} \mathbb{E}_{\pi^{*,k}}\left[\left\langle Q_h^k(s_h, \cdot), \pi_h^{*,k}(\cdot \mid s_h) - \pi_h^k(\cdot \mid s_h)\right\rangle \,\middle|\, s_1 = s_1^k\right]$$

$$+ \sum_{l \in [L]} \sum_{k=(l-1)\tau+1}^{l\tau} \sum_{h \in [H]} \left[\mathbb{E}_{\pi^{*,k}}[\iota_h^k(s_h, a_h) \mid s_1 = s_1^k] - \iota_h^k(s_h^k, a_h^k)\right] + M_{K,H},$$

*where* $M_{K,H} := \sum_{k \in [K]} \sum_{h \in [H]} M_h^k$ *is a martingale that satisfies* $\left|M_h^k\right| \leq 4H$ *for* $(k, h) \in [K] \times [H]$.

We defer its proof to Section B.5.

### B.2    Performance difference bound

We may further decompose the first term on the RHS of Lemma 1 as

$$\sum_{l \in [L]} \sum_{k=(l-1)\tau+1}^{l\tau} \sum_{h \in [H]} \mathbb{E}_{\pi^{*,k}}\left[\left\langle Q_h^k(s_h, \cdot), \pi_h^{*,k}(\cdot \mid s_h) - \pi_h^k(\cdot \mid s_h)\right\rangle \,\middle|\, s_1 = s_1^k\right]$$

$$= \sum_{l\in[L]}\sum_{k=(l-1)\tau+1}^{l\tau}\sum_{h\in[H]}\mathbb{E}_{\pi^*,(l-1)\tau+1}\left[\left.\left\langle Q_h^k(s_h,\cdot),\pi_h^{*,k}(\cdot\mid s_h)-\pi_h^k(\cdot\mid s_h)\right\rangle\,\right|\,s_1=s_1^k\right]$$

$$+\sum_{l\in[L]}\sum_{k=(l-1)\tau+1}^{l\tau}\sum_{h\in[H]}\left(\mathbb{E}_{\pi^*,k}-\mathbb{E}_{\pi^*,(l-1)\tau+1}\right)\left[\left.\left\langle Q_h^k(s_h,\cdot),\pi_h^{*,k}(\cdot\mid s_h)-\pi_h^k(\cdot\mid s_h)\right\rangle\,\right|\,s_1=s_1^k\right].$$

$$(8)$$

### B.2.1 First term in Equation (8)

We first introduce a "one-step descent" result.

**Lemma 2** ([9, Lemma 3.3]). *For any distribution $p^*$ and $p$ supported on $\mathcal{A}$, state $s\in\mathcal{S}$, and function $Q:\mathcal{S}\times\mathcal{A}\to[0,H]$, it holds for a distribution $p'$ supported on $\mathcal{A}$ with $p'(\cdot)\propto p(\cdot)\cdot\exp\{\alpha\cdot Q(s,\cdot)\}$ that*

$$\langle Q(s,\cdot),p^*(\cdot)-p(\cdot)\rangle\le\frac{1}{2}\alpha H^2+\frac{1}{\alpha}\left[D_{KL}(p^*(\cdot)\,\|\,p(\cdot))-D_{KL}(p^*(\cdot)\,\|\,p'(\cdot))\right].$$

The next lemma controls the performance difference for any initial state.

**Lemma 3.** *For any $s_1^k\in\mathcal{S}$, we have*

$$\sum_{l\in[L]}\sum_{k=(l-1)\tau+1}^{l\tau}\sum_{h\in[H]}\mathbb{E}_{\pi^*,(l-1)\tau+1}\left[\left.\left\langle Q_h^k(s_h,\cdot),\pi_h^{*,k}(\cdot\mid s_h)-\pi_h^k(\cdot\mid s_h)\right\rangle\,\right|\,s_1=s_1^k\right]$$

$$\le\frac{1}{2}\alpha KH^3+\frac{1}{\alpha}LH\log A+\tau HP_T.$$

*Proof.* For each $l\in[L]$, we let $\nu^l=\{\nu_h^l\}_{h\in[H]}$ where each $\nu_h^l$ is a policy (or a distribution supported $\mathcal{A}$) to be specified. We have the decomposition

$$\sum_{l\in[L]}\sum_{k=(l-1)\tau+1}^{l\tau}\sum_{h\in[H]}\mathbb{E}_{\pi^*,(l-1)\tau+1}\left[\left.\left\langle Q_h^k(s_h,\cdot),\pi_h^{*,k}(\cdot\mid s_h)-\pi_h^k(\cdot\mid s_h)\right\rangle\,\right|\,s_1=s_1^k\right]$$

$$=\sum_{l\in[L]}\sum_{k=(l-1)\tau+1}^{l\tau}\sum_{h\in[H]}\mathbb{E}_{\pi^*,(l-1)\tau+1}\left[\left.\left\langle Q_h^k(s_h,\cdot),\nu_h^l(\cdot\mid s_h)-\pi_h^k(\cdot\mid s_h)\right\rangle\,\right|\,s_1=s_1^k\right]$$

$$+\sum_{l\in[L]}\sum_{k=(l-1)\tau+1}^{l\tau}\sum_{h\in[H]}\mathbb{E}_{\pi^*,(l-1)\tau+1}\left[\left.\left\langle Q_h^k(s_h,\cdot),\pi_h^{*,k}(\cdot\mid s_h)-\nu_h^l(\cdot\mid s_h)\right\rangle\,\right|\,s_1=s_1^k\right]$$

$$=:E_1+E_2.\qquad(9)$$

By Lemma 2, we have

$$E_1\le\frac{1}{2}\alpha KH^3+\sum_{h\in[H]}\frac{1}{\alpha}$$

$$\times\sum_{l\in[L]}\mathbb{E}_{\pi^*,(l-1)\tau+1}\left[\left.\sum_{k=(l-1)\tau+1}^{l\tau}\left[D_{KL}(\nu_h^l(\cdot\mid s_h)\,\|\,\pi_h^k(\cdot\mid s_h))-D_{KL}(\nu_h^l(\cdot\mid s_h)\,\|\,\pi_h^{k+1}(\cdot\mid s_h))\right]\,\right|\,s_1=s_1^k\right]$$

$$\le\frac{1}{2}\alpha KH^3+\sum_{h\in[H]}\frac{1}{\alpha}$$

$$\times\sum_{l\in[L]}\mathbb{E}_{\pi^*,(l-1)\tau+1}\left[\left.D_{KL}(\nu_h^l(\cdot\mid s_h)\,\|\,\pi_h^{(l-1)\tau+1}(\cdot\mid s_h))-D_{KL}(\nu_h^l(\cdot\mid s_h)\,\|\,\pi_h^{l\tau+1}(\cdot\mid s_h))\,\right|\,s_1=s_1^k\right]$$

$$\le\frac{1}{2}\alpha KH^3+\sum_{h\in[H]}\frac{1}{\alpha}\cdot\sum_{l\in[L]}\mathbb{E}_{\pi^*,(l-1)\tau+1}\left[\left.D_{KL}(\nu_h^l(\cdot\mid s_h)\,\|\,\pi_h^{(l-1)\tau+1}(\cdot\mid s_h))\,\right|\,s_1=s_1^k\right]$$

$$\leq \frac{1}{2}\alpha K H^3 + \frac{1}{\alpha} L H \log A,$$

where the second step holds by telescoping, the third step holds since the KL divergence is non-negative, and the last step holds since by construction $\pi_h^{(l-1)\tau+1}(\cdot \mid s)$ in Algorithm 1 is a uniform distribution on $\mathcal{A}$ and for any policy $\nu$ and state $s \in \mathcal{S}$ we have

$$D_{\mathrm{KL}}(\nu(\cdot \mid s)\|\pi_h^{(l-1)\tau+1}(\cdot \mid s)) = \sum_{a \in \mathcal{A}} \nu(a \mid s) \cdot \log\left(A \cdot \nu(a \mid s)\right)$$

$$= \log A + \sum_{a \in \mathcal{A}} \nu(a \mid s) \cdot \log\left(\nu(a \mid s)\right)$$

$$\leq \log A$$

given the fact that the entropy of any distribution is non-negative.

Now for each $(l, h) \in [L] \times [H]$, we set

$$\nu_h^l := \pi_h^{*,(l-1)\tau+1},$$

that is, $\nu_h^l$ is the policy after one update in step $h$ of period $l$. For $D_2$, we have

$$E_2 \leq \sum_{l \in [L]} \sum_{k=(l-1)\tau+1}^{l\tau} \sum_{h \in [H]} \mathbb{E}_{\pi^{*,(l-1)\tau+1}}\left[ H \cdot \|\pi_h^{*,k}(\cdot \mid s_h) - \nu_h^l(\cdot \mid s_h)\|_1 \,\Big|\, s_1 = s_1^k \right]$$

$$= H \cdot \sum_{l \in [L]} \sum_{k=(l-1)\tau+1}^{l\tau} \sum_{h \in [H]} \mathbb{E}_{\pi^{*,(l-1)\tau+1}}\left[ \|\pi_h^{*,k}(\cdot \mid s_h) - \pi_h^{*,(l-1)\tau+1}(\cdot \mid s_h)\|_1 \,\Big|\, s_1 = s_1^k \right]$$

$$\leq H \cdot \sum_{l \in [L]} \sum_{k=(l-1)\tau+1}^{l\tau} \sum_{h \in [H]} \sum_{t=(l-1)\tau+2}^{k} \mathbb{E}_{\pi^{*,(l-1)\tau+1}}\left[ \|\pi_h^{*,t}(\cdot \mid s_h) - \pi_h^{*,t-1}(\cdot \mid s_h)\|_1 \,\Big|\, s_1 = s_1^k \right]$$

$$\leq H \cdot \sum_{l \in [L]} \sum_{k=(l-1)\tau+1}^{l\tau} \sum_{t=(l-1)\tau+1}^{l\tau} \sum_{h \in [H]} \max_{s' \in \mathcal{S}} \|\pi_h^{*,t}(\cdot \mid s') - \pi_h^{*,t-1}(\cdot \mid s')\|_1$$

$$= H \cdot \tau \cdot \sum_{t \in [K]} \sum_{h \in [H]} \max_{s' \in \mathcal{S}} \|\pi_h^{*,t}(\cdot \mid s') - \pi_h^{*,t-1}(\cdot \mid s')\|_1$$

$$= H \cdot \tau \cdot P_T$$

where the first step holds by Holder's inequality and the fact that $\|Q_h^k(s, \cdot)\|_\infty \leq H$, the second step holds by the definition of $\{\nu_h^l\}$, the third step follows from telescoping, and the last step holds by the definition $P_T := \sum_{k \in [K]} \sum_{h \in [H]} \|\pi_h^{*,k} - \pi_h^{*,k-1}\|_\infty$. $\qquad \square$

### B.2.2 Second term in Equation (8)

The following lemma controls the performance difference due to varying optimal policies across episodes.

**Lemma 4.** *Under Assumption 1, we have*

$$\sum_{l \in [L]} \sum_{k=(l-1)\tau+1}^{l\tau} \sum_{h \in [H]} (\mathbb{E}_{\pi^{*,k}} - \mathbb{E}_{\pi^{*,(l-1)\tau+1}})\left[ \left\langle Q_h^k(s_h, \cdot), \pi_h^{*,k}(\cdot \mid s_h) - \pi_h^k(\cdot \mid s_h)\right\rangle \,\Big|\, s_1 = s_1^k \right]$$

$$\leq C \cdot \tau H^2 P_T,$$

*where $C > 0$ is a universal constant.*

*Proof.* We denote by $\mathbb{I}(s_h)$ the indicator function for state $s_h$, and we have

$$\sum_{l \in [L]} \sum_{k=(l-1)\tau+1}^{l\tau} \sum_{h \in [H]} (\mathbb{E}_{\pi^{*,k}} - \mathbb{E}_{\pi^{*,(l-1)\tau+1}})\left[ \left\langle Q_h^k(s_h, \cdot), \pi_h^{*,k}(\cdot \mid s_h) - \pi_h^k(\cdot \mid s_h)\right\rangle \,\Big|\, s_1 = s_1^k \right]$$

$$\leq \sum_{l\in[L]}\sum_{k=(l-1)\tau+1}^{l\tau}\sum_{h\in[H]}(\mathbb{E}_{\pi^*,k}-\mathbb{E}_{\pi^*,(l-1)\tau+1})\Big[2H\cdot\mathbb{I}(s_h)\ \Big|\ s_1=s_1^k\Big]$$

$$=\sum_{l\in[L]}\sum_{k=(l-1)\tau+1}^{l\tau}\sum_{h\in[H]}\sum_{t=(l-1)\tau+2}^{k}(\mathbb{E}_{\pi^*,t}-\mathbb{E}_{\pi^*,t-1})\Big[2H\cdot\mathbb{I}(s_h)\ \Big|\ s_1=s_1^k\Big] \tag{10}$$

where the first step follows from $\left|\left\langle Q_h^k(s_h,\cdot),\pi_h^{*,k}(\cdot\mid s_h)-\pi_h^k(\cdot\mid s_h)\right\rangle\right|\leq 2H\cdot\mathbb{I}(s_h)$ and the last step holds by telescoping. Let $\mathcal{P}_i^\pi(s)$ be the visitation measure of state $s$ in step $i$ under policy $\pi$, and let us fix an $h\in[H]$. Under policies $\{\pi^{(i)}\}$, the distribution of $s_h$ conditional on $s_1$ is given by

$$\mathcal{P}_1^{\pi^{(1)}}\mathcal{P}_2^{\pi^{(2)}}\cdots\mathcal{P}_{h-1}^{\pi^{(h-1)}}(s_h\mid s_1):=\sum_{s_2,\ldots,s_{h-1}}\prod_{i\in[h-1]}\mathcal{P}_i^{\pi^{(i)}}(s_{i+1}\mid s_i).$$

Recall that $\|\pi-\pi'\|_\infty:=\max_{s\in\mathcal{S}}\|\pi(\cdot\mid s)-\pi'(\cdot\mid s)\|_1$ for any pair of policies $\pi$ and $\pi'$, and $\mathcal{P}_h^\pi(s\mid s'):=\sum_{a'\in\mathcal{A}}\mathcal{P}_h(s\mid s',a')\cdot\pi_h(a'\mid s')$ is the transition kernel in step $h$ when policy $\pi$ is executed. We have the following smoothness property for the (conditional) visitation measure $\mathcal{P}_1^{\pi^{(1)}}\mathcal{P}_2^{\pi^{(2)}}\cdots\mathcal{P}_{h-1}^{\pi^{(h-1)}}(s_h\mid s_1)$ thanks to Assumption 1.

**Lemma 5.** *Under Assumption 1, for any $h\in[H]$, $j\in[h-1]$, $s_h,s_1\in\mathcal{S}$, and policies $\{\pi^{(i)}\}_{i\in[H]}\cup\{\pi'\}$ we have*

$$\left|\mathcal{P}_1^{\pi^{(1)}}\cdots\mathcal{P}_j^{\pi^{(j)}}\cdots\mathcal{P}_{h-1}^{\pi^{(h-1)}}(s_h\mid s_1)-\mathcal{P}_1^{\pi^{(1)}}\cdots\mathcal{P}_j^{\pi'}\cdots\mathcal{P}_{h-1}^{\pi^{(h-1)}}(s_h\mid s_1)\right|$$
$$\leq C\cdot\|\pi_j^{(j)}-\pi_j'\|_\infty,$$

*where $C>0$ is a universal constant.*

*Proof.* We have

$$\left|\mathcal{P}_1^{\pi^{(1)}}\cdots\mathcal{P}_j^{\pi^{(j)}}\cdots\mathcal{P}_{h-1}^{\pi^{(h-1)}}(s_h\mid s_1)-\mathcal{P}_1^{\pi^{(1)}}\cdots\mathcal{P}_j^{\pi'}\cdots\mathcal{P}_{h-1}^{\pi^{(h-1)}}(s_h\mid s_1)\right|$$

$$\leq\sum_{s_2,s_3,\ldots,s_{h-1}}\left|\mathcal{P}_j^{\pi^{(j)}}(s_{j+1}\mid s_j)-\mathcal{P}_j^{\pi'}(s_{j+1}\mid s_j)\right|\cdot\prod_{i\in[h-1]\backslash\{j\}}\mathcal{P}_i^{\pi^{(i)}}(s_{i+1}\mid s_i)$$

$$\overset{(i)}{\leq}\sum_{s_2,\ldots s_j,s_{j+2},\ldots,s_{h-1}}\sum_{s_{j+1}}\left|\mathcal{P}_j^{\pi^{(j)}}(s_{j+1}\mid s_j)-\mathcal{P}_j^{\pi'}(s_{j+1}\mid s_j)\right|\cdot\max_{s_{j+1}\in\mathcal{S}}\prod_{i\in[h-1]\backslash\{j\}}\mathcal{P}_i^{\pi^{(i)}}(s_{i+1}\mid s_i)$$

$$\overset{(ii)}{\leq}\sum_{s_2,\ldots s_{j-1},s_{j+2},\ldots,s_{h-1}}\max_{s_j\in\mathcal{S}}\sum_{s_{j+1}}\left|\mathcal{P}_j^{\pi^{(j)}}(s_{j+1}\mid s_j)-\mathcal{P}_j^{\pi'}(s_{j+1}\mid s_j)\right|\cdot\sum_{s_j}\max_{s_{j+1}\in\mathcal{S}}\prod_{i\in[h-1]\backslash\{j\}}\mathcal{P}_i^{\pi^{(i)}}(s_{i+1}\mid s_i)$$

$$\overset{(iii)}{\leq}C\cdot\|\pi_j^{(j)}-\pi_j'\|_\infty\cdot\sum_{s_2,\ldots,s_j,s_{j+2}\ldots,s_{h-1}}\max_{s_{j+1}\in\mathcal{S}}\prod_{i\in[h-1]\backslash\{j\}}\mathcal{P}_i^{\pi^{(i)}}(s_{i+1}\mid s_i)$$

$$=C\cdot\|\pi_j^{(j)}-\pi_j'\|_\infty\cdot\underbrace{\sum_{s_{j+2},\ldots,s_{h-1}}\max_{s_{j+1}\in\mathcal{S}}\prod_{i=j+1}^{h-1}\mathcal{P}_i^{\pi^{(i)}}(s_{i+1}\mid s_i)}_{\leq1}\cdot\underbrace{\sum_{s_2,\ldots,s_j}\prod_{i=1}^{j-1}\mathcal{P}_i^{\pi^{(i)}}(s_{i+1}\mid s_i)}_{=1}$$

$$\leq C\cdot\|\pi_j^{(j)}-\pi_j'\|_\infty,$$

where steps $(i)$ and $(ii)$ hold by Holder's inequality, and step $(iii)$ holds under Assumption 1. $\qquad\square$

Therefore, for $(k,t,h)\in[K]^2\times[H]$ such that $k\leq t-1$, we have

$$\left|(\mathbb{E}_{\pi^*,t}-\mathbb{E}_{\pi^*,t-1})\Big[\mathbb{I}(s_h)\ \Big|\ s_1=s_1^k\Big]\right|$$
$$\leq\|\mathcal{P}_1^{\pi^{*,t}}\mathcal{P}_2^{\pi^{*,t}}\cdots\mathcal{P}_{h-1}^{\pi^{*,t}}(\cdot\mid s_1^k)-\mathcal{P}_1^{\pi^{*,t-1}}\mathcal{P}_2^{\pi^{*,t-1}}\cdots\mathcal{P}_{h-1}^{\pi^{*,t-1}}(\cdot\mid s_1^k)\|_\infty$$
$$\leq\|\mathcal{P}_1^{\pi^{*,t}}\mathcal{P}_2^{\pi^{*,t}}\cdots\mathcal{P}_{h-1}^{\pi^{*,t}}(\cdot\mid s_1^k)-\mathcal{P}_1^{\pi^{*,t}}\mathcal{P}_2^{\pi^{*,t-1}}\cdots\mathcal{P}_{h-1}^{\pi^{*,t-1}}(\cdot\mid s_1^k)\|_\infty$$

$$+ \|\mathcal{P}_1^{\pi^{*,t}}\mathcal{P}_2^{\pi^{*,t-1}} \cdots \mathcal{P}_{h-1}^{\pi^{*,t-1}}(\cdot \mid s_1^k) - \mathcal{P}_1^{\pi^{*,t-1}}\mathcal{P}_2^{\pi^{*,t-1}} \cdots \mathcal{P}_{h-1}^{\pi^{*,t-1}}(\cdot \mid s_1^k)\|_\infty$$

$$\leq C \cdot \sum_{i \in [h]} \|\pi_i^{*,t} - \pi_i^{*,t-1}\|_\infty, \tag{11}$$

where the third step follows from further telescoping the first term in the second step and then applying Lemma 5. Combining Equations (10) and (11), we have

$$\sum_{l \in [L]} \sum_{k=(l-1)\tau+1}^{l\tau} \sum_{h \in [H]} (\mathbb{E}_{\pi^{*,k}} - \mathbb{E}_{\pi^{*,(l-1)\tau+1}}) \left[ \left\langle Q_h^k(s_h, \cdot), \pi_h^{*,k}(\cdot \mid s_h) - \pi_h^k(\cdot \mid s_h) \right\rangle \,\middle|\, s_1 = s_1^k \right]$$

$$\leq \sum_{l \in [L]} \sum_{k=(l-1)\tau+1}^{l\tau} \sum_{h \in [H]} \sum_{t=(l-1)\tau+2}^{k} 2H \cdot C \cdot \sum_{i \in [h]} \|\pi_i^{*,t} - \pi_i^{*,t-1}\|_\infty$$

$$\leq 2H \cdot C \cdot \sum_{h \in [H]} \left( \sum_{l \in [L]} \sum_{k=(l-1)\tau+1}^{l\tau} \sum_{t=(l-1)\tau+1}^{l\tau} \sum_{i \in [H]} \|\pi_i^{*,t} - \pi_i^{*,t-1}\|_\infty \right)$$

$$= 2H \cdot C \cdot \sum_{h \in [H]} \left( \tau \sum_{t \in [K]} \sum_{i \in [H]} \|\pi_i^{*,t} - \pi_i^{*,t-1}\|_\infty \right)$$

$$\leq 2C \cdot H^2 \cdot \tau \cdot P_T,$$

where in the last step we used the definition $P_T := \sum_{k \in [K]} \sum_{i \in [H]} \|\pi_i^{*,k} - \pi_i^{*,k-1}\|_\infty$. $\qquad\square$

### B.2.3 Putting together

Finally, we establish the following result on the performance difference.

**Lemma 6.** *Recall that* $P_T := \sum_{k \in [K]} \sum_{i \in [H]} \|\pi_i^{*,k} - \pi_i^{*,k-1}\|_\infty$. *Under Assumption 1, we choose* $\alpha = \sqrt{\frac{L \log A}{KH^2}}$ *in Algorithm 1, and we have*

$$\sum_{l \in [L]} \sum_{k=(l-1)\tau+1}^{l\tau} \sum_{h \in [H]} \mathbb{E}_{\pi^{*,k}} \left[ \left\langle Q_h^k(s_h, \cdot), \pi_h^{*,k}(\cdot \mid s_h) - \pi_h^k(\cdot \mid s_h) \right\rangle \,\middle|\, s_1 = s_1^k \right]$$

$$= 2H^2\sqrt{K \log A} + C \cdot \tau H^2 P_T,$$

*for some universal constant* $C > 0$.

*Proof.* Recall from Equation (8) that for any $l \in [L]$, we have

$$\sum_{l \in [L]} \sum_{k=(l-1)\tau+1}^{l\tau} \sum_{h \in [H]} \mathbb{E}_{\pi^{*,k}} \left[ \left\langle Q_h^k(s_h, \cdot), \pi_h^{*,k}(\cdot \mid s_h) - \pi_h^k(\cdot \mid s_h) \right\rangle \,\middle|\, s_1 = s_1^k \right]$$

$$= \sum_{l \in [L]} \sum_{k=(l-1)\tau+1}^{l\tau} \sum_{h \in [H]} \mathbb{E}_{\pi^{*,(l-1)\tau+1}} \left[ \left\langle Q_h^k(s_h, \cdot), \pi_h^{*,k}(\cdot \mid s_h) - \pi_h^k(\cdot \mid s_h) \right\rangle \,\middle|\, s_1 = s_1^k \right]$$

$$+ \sum_{l \in [L]} \sum_{k=(l-1)\tau+1}^{l\tau} \sum_{h \in [H]} (\mathbb{E}_{\pi^{*,k}} - \mathbb{E}_{\pi^{*,(l-1)\tau+1}}) \left[ \left\langle Q_h^k(s_h, \cdot), \pi_h^{*,k}(\cdot \mid s_h) - \pi_h^k(\cdot \mid s_h) \right\rangle \,\middle|\, s_1 = s_1^k \right].$$

By applying Lemmas 3 and 4, we have

$$\sum_{l \in [L]} \sum_{k=(l-1)\tau+1}^{l\tau} \sum_{h \in [H]} \mathbb{E}_{\pi^{*,k}} \left[ \left\langle Q_h^k(s_h, \cdot), \pi_h^{*,k}(\cdot \mid s_h) - \pi_h^k(\cdot \mid s_h) \right\rangle \,\middle|\, s_1 = s_1^k \right]$$

$$\leq \alpha K H^3 + \frac{1}{\alpha} L H \log A + \tau H P_T + C' \cdot \tau H^2 P_T$$

$$= 2H^2\sqrt{KL \log A} + \tau H P_T + C' \cdot \tau H^2 P_T$$

$$\leq 2H^2\sqrt{KL \log A} + C \cdot \tau H^2 P_T,$$

where the equality above holds by our choice of $\alpha$, and $C, C' > 0$ are universal constants. $\qquad\square$

## B.3 Model prediction error

We need the following results to control the bonus $\Gamma_h^k(\cdot, \cdot)$ (defined in Line 5 of Algorithm 3) accumulated over episodes.

**Lemma 7.** *Let $\lambda = 1$ and $\beta = C \cdot H \sqrt{S \log(dT/p)}$ in Algorithm 1, where $C > 0$ is a universal constant and $p \in (0, 1]$. With probability at least $1 - p/2$ and for all $(k, h, s, a) \in [K] \times [H] \times \mathcal{S} \times \mathcal{A}$, it holds that*

$$-2\Gamma_h^k(s, a) \leq \iota_h^k(s, a) \leq 0.$$

*Proof.* The proof follows that of [9, Lemma 4.3] specialized to the tabular setting by replacing Lemma D.2 therein with [5, Lemma 12]. $\qquad\square$

**Lemma 8** ([9, Lemma D.6]; [29, Lemma D.2]). *Let $\{\phi_t\}_{t \geq 0}$ be a bounded sequence in $\mathbb{R}^d$ satisfying $\sup_{t \geq 0} \|\phi_t\| \leq 1$. Let $\Lambda_0 \in \mathbb{R}^{d \times d}$ be a positive definite matrix with $\lambda_{\min}(\Lambda_0) \geq 1$. For any $t \geq 0$, we define $\Lambda_t := \Lambda_0 + \sum_{i \in [t-1]} \phi_i \phi_i^\top$. Then, we have*

$$\log\left[\frac{\det(\Lambda_{t+1})}{\det(\Lambda_0)}\right] \leq \sum_{i \in [t]} \phi_i^\top \Lambda_i^{-1} \phi_i \leq 2\log\left[\frac{\det(\Lambda_{t+1})}{\det(\Lambda_0)}\right].$$

**Lemma 9.** *We have*

$$\sum_{k \in [K[} \sum_{h \in [H]} \Gamma_h^k(s_h^k, a_h^k) \leq \beta H \sqrt{2dK \log((K + \lambda)/\lambda)}.$$

*Proof.* Given the construction of $\Lambda_h^k$ in Algorithm 1, we have for any $h \in [H]$,

$$\sum_{k \in [K]} \phi(s_h^k, a_h^k)^\top (\Lambda_h^k)^{-1} \phi(s_h^k, a_h^k) \leq 2\log\left[\frac{\det(\Lambda_h^{K+1})}{\det(\Lambda_h^1)}\right]$$

$$\leq 2d\log\left[\frac{K + \lambda}{\lambda}\right],$$

where the last step holds since the construction of Algorithm 1 implies that $\Lambda_h^1 = \lambda \cdot \mathbf{I}$ and

$$\Lambda_h^{k+1} = \sum_{t \in [k]} \phi(s_h^t, a_h^t) \phi(s_h^t, a_h^t)^\top + \lambda \cdot \mathbf{I} \preceq (k + \lambda) \cdot \mathbf{I},$$

which yields

$$\log\left[\frac{\det(\Lambda_h^{K+1})}{\det(\Lambda_h^1)}\right] \leq \log\left[\frac{\det((K + \lambda) \cdot \mathbf{I})}{\det(\lambda \cdot \mathbf{I})}\right] = d\log\left[\frac{K + \lambda}{\lambda}\right].$$

Therefore, by the Cauchy-Schwarz inequality and Lemma 9, we have

$$\sum_{k \in [K[} \sum_{h \in [H]} \Gamma_h^k(s_h^k, a_h^k) \leq \beta \cdot \sum_{h \in [H]} \left(K \cdot \sum_{k \in [K]} \phi(s_h^k, a_h^k)^\top (\Lambda_h^k)^{-1} \phi(s_h^k, a_h^k)\right)^{1/2}$$

$$= \beta H \sqrt{2dK \log((K + \lambda)/\lambda)}.$$

$\qquad\square$

## B.4 Martingale bound

**Lemma 10.** *Consider $M_{K,H}$ in Lemma 1. With probability $1 - \delta/2$, we have*

$$|M_{K,H}| \leq \sqrt{16H^2 T \cdot \log(4/\delta)}.$$

*Proof.* From Lemma 1 and by the Azuma Hoeffding inequality, we have for any $t \geq 0$,

$$\mathbb{P}\left(|M_{K,H}| \geq t\right) \leq 2\exp\left(-\frac{t^2}{16H^2 T}\right).$$

Setting $t = \sqrt{16H^2 T \cdot \log(4/\delta)}$, we have

$$|M_{K,H}| \leq \sqrt{16H^2 T \cdot \log(4/\delta)}$$

with probability at least $1 - \delta/2$. $\qquad\square$

## B.5 Proof of Lemma 1

For any function $f : \mathcal{S} \times \mathcal{A} \to \mathbb{R}$ and any $(k, h, s) \in [K] \times [H] \times \mathcal{S}$, define the operators

$$(\mathbb{J}^*_{k,h} f)(s) = \left\langle f(s, \cdot), \pi_h^{*,k}(\cdot \mid s) \right\rangle, \qquad (\mathbb{J}_{k,h} f)(s) = \left\langle f(s, \cdot), \pi_h^k(\cdot \mid s) \right\rangle.$$

and the function

$$\xi_h^k(s) := (\mathbb{J}^*_{k,h} Q_h^k)(s) - (\mathbb{J}_{k,h} Q_h^k)(s) = \left\langle Q_h^k(s, \cdot), \pi_h^{*,k}(\cdot \mid s) - \pi_h^k(\cdot \mid s) \right\rangle.$$

The proof mostly follows that of [9, Lemma 4.2], except that we replace $\pi^*$ and $\mathbb{J}_h$ therein by $\pi^{*,k}$ and $\mathbb{J}^*_{k,h}$, respectively. Therefore, we outline the key steps only and refer the readers to the proof of [9, Lemma 4.2] for full details.

Recall that $\pi^{*,k}$ is the optimal policy in episode $k$. We have

$$\text{D-Regret}(K) = \sum_{k \in [K]} \left[ V_1^{\pi^{*,k},k}(s_1^k) - V_1^{\pi^k,k}(s_1^k) \right]$$

$$= \sum_{l \in [L]} \sum_{k=(l-1)\tau+1}^{l\tau} \left[ V_1^{\pi^{*,k},k}(s_1^k) - V_1^{\pi^k,k}(s_1^k) \right].$$

We have

$$V_1^{\pi^{*,k},k}(s_1^k) - V_1^{\pi^k,k}(s_1^k) = \underbrace{V_1^{\pi^{*,k},k}(s_1^k) - V_1^k(s_1^k)}_{G_1} + \underbrace{V_1^k(s_1^k) - V_1^{\pi^k,k}(s_1^k)}_{G_2}. \tag{12}$$

From [9, Section B.1], we have for any $k \in [K]$,

$$G_1 = \sum_{h \in [H]} \mathbb{E}_{\pi^{*,k}}[\iota_h^k(s_h, a_h) \mid s_1 = s_1^k]$$

$$+ \sum_{h \in [H]} \mathbb{E}_{\pi^{*,k}} \left[ \left\langle Q_h^k(s_h, \cdot), \pi_h^{*,k}(\cdot \mid s_h) - \pi_h^k(\cdot \mid s_h) \right\rangle \,\middle|\, s_1 = s_1^k \right], \tag{13}$$

and

$$G_2 = -\sum_{h \in [H]} \iota_h^k(s_h^k, a_h^k) + \sum_{h \in [H]} (D_{h,1}^k + D_{h,2}^k), \tag{14}$$

where

$$D_{h,1}^k := \left( \mathbb{J}_{k,h}(Q_h^k - Q_h^{\pi^k,k}) \right)(s_h^k) - (Q_h^k - Q_h^{\pi^k,k})(s_h^k, a_h^k),$$

$$D_{h,2}^k := \left( \mathbb{P}_h(V_{h+1}^k - V_{h+1}^{\pi^k,k}) \right)(s_h^k, a_h^k) - (V_{h+1}^k - V_{h+1}^{\pi^k,k})(s_{h+1}^k).$$

From Line 11 of Algorithm 1, we have

$$Q_h^k, Q_h^{\pi^k,k}, V_{h+1}^k, V_{h+1}^{\pi^k,k} \in [0, H],$$

which implies $\left| D_{h,1}^k \right|, \left| D_{h,2}^k \right| \le 2H$ for any $(k, h) \in [K] \times [H]$. Writing $M_h^k := D_{h,1}^k + D_{h,2}^k$, we have that

$$M_{K,H} := \sum_{k \in [K]} \sum_{h \in [H]} M_h^k$$

is a martingale where $\left| M_h^k \right| \le 4H$. The proof is completed in view of Equations (12), (13) and (14).

## C  Proof of Theorem 1

By Lemmas 7 and 9, we have

$$\sum_{l\in[L]}\sum_{k=(l-1)\tau+1}^{l\tau}\sum_{h\in[H]}\left[\mathbb{E}_{\pi^{*,k}}[\iota_h^k(s_h,a_h)\mid s_1=s_1^k]-\iota_h^k(s_h^k,a_h^k)\right]$$

$$\leq 2\sum_{k\in[K[}\sum_{h\in[H]}\Gamma_h^k(s_h^k,a_h^k)\leq 2\beta H\sqrt{2dK\log((K+\lambda)/\lambda)}. \tag{15}$$

We apply Lemmas 6 and 10 as well as Equation (15) to the conclusion of Lemma 1. With the choice of $\lambda=1$ and $\beta=C_\beta H\sqrt{S\log(dT/\delta)}$ and the identity $K=L\tau$, we have

$$\text{D-Regret}(K)\leq 2H^2\sqrt{KL\log A}+C\cdot\tau H^2 P_T+2\beta H\sqrt{2dK\log((K+\lambda)/\lambda)}$$

$$+\sqrt{16H^2 T\cdot\log(4/\delta)}$$

$$\leq 2H^2\sqrt{KL\log A}+C\cdot\tau H^2 P_T+2C_\beta H^2\sqrt{S\log(dT/\delta)}\sqrt{2dK\log(K+1)}$$

$$+\sqrt{16H^2 T\cdot\log(4/\delta)}$$

$$= 2H^2\sqrt{KL\log A}+C\cdot\tau H^2 P_T+2C_\beta\sqrt{2H^3 S^2 AT\cdot\log(dT/\delta)\cdot\log(K+1)}$$

$$+\sqrt{16H^2 T\cdot\log(4/\delta)}$$

$$\leq 2H^2\sqrt{KL\log A}+C\cdot\tau H^2 P_T+C'\sqrt{2H^3 S^2 AT\cdot\log^2(dT/\delta)} \tag{16}$$

where $C,C'>0$ are universal constants, the second step above holds by the definition of $\beta$, and the third step holds by the identity $T=KH$.

We discuss several cases.

- If $0\leq P_T\leq\sqrt{\frac{\log A}{K}}$, then by elementary calculation we have $\left(\frac{T\sqrt{\log A}}{HP_T}\right)^{2/3}\geq K$. This implies $\tau=K$ by our choice of $\tau$, and therefore $L=1$. Then Equation (16) yields

$$\text{D-Regret}(K)\leq 2H^2\sqrt{K\log A}+C\cdot H^2\sqrt{K\log A}+C'\sqrt{2H^3 S^2 AT\log^2(dT/\delta)}$$

$$=(2+C)\sqrt{H^3 T\log A}+C'\sqrt{2H^3 S^2 AT\log^2(dT/\delta)}.$$

- If $\sqrt{\frac{\log A}{K}}\leq P_T\leq 2^{-3/2}\cdot K\sqrt{\log A}$, we have and $2\leq\tau\leq K$ and Equation (16) implies

$$\text{D-Regret}(K)\leq 2\cdot\frac{1}{\sqrt{\tau}}HT\sqrt{\log A}+C\cdot\tau H^2 P_T+C'\sqrt{2H^3 S^2 AT\cdot\log^2(dT/\delta)}$$

$$\leq(4+C)\cdot\left(H^2 T\sqrt{\log A}\right)^{2/3}P_T^{1/3}+C'\sqrt{2H^3 S^2 AT\cdot\log^2(dT/\delta)},$$

where the first step holds by $K=L\tau$, and in the last step we applied the choice of $\tau=\left\lfloor\left(\frac{T\sqrt{\log A}}{HP_T}\right)^{2/3}\right\rfloor$.

- If $P_T>2^{-3/2}\cdot K\sqrt{\log A}$, we have $\left(\frac{T\sqrt{\log A}}{HP_T}\right)^{2/3}<2$ and therefore $\tau=1$ and $L=K$. Then Equation (16) yields

$$\text{D-Regret}(K)\leq 2HT\sqrt{\log A}+C\cdot H^2 P_T+C'\sqrt{2H^3 S^2 AT\cdot\log^2(dT/\delta)}$$

$$\leq(8+C)H^2 P_T+C'\sqrt{2H^3 S^2 AT\cdot\log^2(dT/\delta)},$$

It is not hard to see that all of the above arguments also go through if we replace $P_T$ with its upper bound. The proof is completed by combining the last case above with the trivial bound $\text{D-Regret}(K)\leq T$.

## D  Proof of Theorem 2

The proof follows the same reasoning as in Appendix C, except that Lemmas 6 no longer applies. In the following, we provide an alternative to Lemmas 6 adapted for Algorithm 2.

**Lemma 11.** *For any $s \in \mathcal{S}$, we have*

$$\sum_{l \in [L]} \sum_{k=(l-1)\tau+1}^{l\tau} \sum_{h \in [H]} \left\langle Q_h^k(s, \cdot), \pi_h^{*,k}(\cdot \mid s) - \pi_h^k(\cdot \mid s) \right\rangle$$

$$\leq \alpha D_T + \frac{1}{\alpha} LH \log A + \tau H P_T.$$

*Proof.* Let us fix an $s \in \mathcal{S}$. For each $l \in [L]$, we let $\nu^l = \{\nu_h^l\}_{h \in [H]}$ where each $\nu_h^l$ is a policy (or a distribution supported $\mathcal{A}$) that depends only on $l$ and $h$. We have the decomposition

$$\sum_{l \in [L]} \sum_{k=(l-1)\tau+1}^{l\tau} \sum_{h \in [H]} \left\langle Q_h^k(s, \cdot), \pi_h^{*,k}(\cdot \mid s) - \pi_h^k(\cdot \mid s) \right\rangle$$

$$\leq \sum_{l \in [L]} \sum_{k=(l-1)\tau+1}^{l\tau} \sum_{h \in [H]} \left\langle Q_h^k(s, \cdot), \nu_h^l(\cdot \mid s) - \pi_h^k(\cdot \mid s) \right\rangle$$

$$+ \sum_{l \in [L]} \sum_{k=(l-1)\tau+1}^{l\tau} \sum_{h \in [H]} \left\langle Q_h^k(s, \cdot), \pi_h^{*,k}(\cdot \mid s) - \nu_h^l(\cdot \mid s) \right\rangle$$

$$=: E_1 + E_2.$$

The term $E_2$ can be controlled in exactly the same way as in the proof of Lemma 3. Therefore, we only control $E_1$. Note that the policy update steps in Algorithm 2 (Lines 8 and 12) essentially follow the update steps of OMD (see e.g. [50, Section 3.1.1] for details). This observation enables us to take advantage of the following lemma, which is a version of [50, Proposition 5] adapted to our case.

**Lemma 12.** *For any $(l, h, s) \in [L] \times [H] \times \mathcal{S}$, we have*

$$\sum_{k=(l-1)\tau+1}^{l\tau} \left\langle Q_h^k(s, \cdot), \nu_h^l(\cdot \mid s) - \pi_h^k(\cdot \mid s) \right\rangle$$

$$\leq \frac{\log A}{\alpha} + \alpha \cdot \sum_{k=(l-1)\tau+1}^{l\tau} \|Q_h^k(s, \cdot) - Q_h^{k-1}(s, \cdot)\|_\infty^2 - \frac{1}{8\alpha} \cdot \sum_{k=(l-1)\tau+1}^{l\tau} \|\pi_h^k - \pi_h^{k-1}\|_\infty^2.$$

*Proof.* The result follows from [50, Proposition 5] and we note that the quantity $R$ defined therein is upper bounded by $\log A$. $\qquad\square$

By Lemma 12 and the definition of $D_T$ in Equation (5), we have

$$E_1 \leq L \cdot H \cdot \frac{\log A}{\alpha} + \alpha \cdot D_T,$$

We have the following result on the performance difference, similar to Lemma 6. $\qquad\square$

**Lemma 13.** *Recall that $P_T := \sum_{k \in [K]} \sum_{i \in [H]} \|\pi_i^{*,k} - \pi_i^{*,k-1}\|_\infty$. Under Assumption 1, we choose $\alpha = \sqrt{\frac{LH \log A}{D_T}}$ in Algorithm 2, and we have*

$$\sum_{l \in [L]} \sum_{k=(l-1)\tau+1}^{l\tau} \sum_{h \in [H]} \mathbb{E}_{\pi^{*,k}} \left[ \left\langle Q_h^k(s_h, \cdot), \pi_h^{*,k}(\cdot \mid s_h) - \pi_h^k(\cdot \mid s_h) \right\rangle \;\middle|\; s_1 = s_1^k \right]$$

$$= 2\sqrt{D_T LH \log A} + C \cdot \tau H^2 P_T,$$

*for some universal constant $C > 0$.*

*Proof.* Now, for any $l \in [L]$ we have the decomposition

$$\sum_{l \in [L]} \sum_{k=(l-1)\tau+1}^{l\tau} \sum_{h \in [H]} \mathbb{E}_{\pi^{*,k}} \left[ \left\langle Q_h^k(s_h, \cdot), \pi_h^{*,k}(\cdot \mid s_h) - \pi_h^k(\cdot \mid s_h) \right\rangle \mid s_1 = s_1^k \right]$$

$$= \sum_{l \in [L]} \sum_{k=(l-1)\tau+1}^{l\tau} \sum_{h \in [H]} \mathbb{E}_{\pi^{*,(l-1)\tau+1}} \left[ \left\langle Q_h^k(s_h, \cdot), \pi_h^{*,k}(\cdot \mid s_h) - \pi_h^k(\cdot \mid s_h) \right\rangle \mid s_1 = s_1^k \right]$$

$$+ \sum_{l \in [L]} \sum_{k=(l-1)\tau+1}^{l\tau} \sum_{h \in [H]} (\mathbb{E}_{\pi^{*,k}} - \mathbb{E}_{\pi^{*,(l-1)\tau+1}}) \left[ \left\langle Q_h^k(s_h, \cdot), \pi_h^{*,k}(\cdot \mid s_h) - \pi_h^k(\cdot \mid s_h) \right\rangle \mid s_1 = s_1^k \right]$$

By applying Lemmas 11 and 4, we have

$$\sum_{l \in [L]} \sum_{k=(l-1)\tau+1}^{l\tau} \sum_{h \in [H]} \mathbb{E}_{\pi^{*,k}} \left[ \left\langle Q_h^k(s_h, \cdot), \pi_h^{*,k}(\cdot \mid s_h) - \pi_h^k(\cdot \mid s_h) \right\rangle \mid s_1 = s_1^k \right]$$

$$\leq \alpha D_T + \frac{LH \log A}{\alpha} + \tau H P_T + C' \cdot \tau H^2 P_T$$

$$= 2\sqrt{D_T LH \log A} + \tau H P_T + C' \cdot \tau H^2 P_T$$

$$\leq 2\sqrt{D_T LH \log A} + C \cdot \tau H^2 P_T,$$

where the last equality holds by our choice of $\alpha$, and $C, C' > 0$ are universal constants $\qquad \square$

We apply Lemmas 13 and 10 and Equation (15) to the conclusion of Lemma 1. With the choice of $\lambda = 1$ and $\beta = C_\beta H \sqrt{S \log(dT/\delta)}$, we have

$$\begin{aligned}
\text{D-Regret}(K) &\leq 2\sqrt{D_T LH \log A} + C \cdot \tau H^2 P_T + 2\beta H \sqrt{2dK \log((K+\lambda)/\lambda)} \\
&\quad + \sqrt{16H^2 T \cdot \log(4/\delta)} \\
&\leq 2\sqrt{D_T LH \log A} + C \cdot \tau H^2 P_T + 2C_\beta H^2 \sqrt{S \log(dT/\delta)} \sqrt{2dK \log(K+1)} \\
&\quad + \sqrt{16H^2 T \cdot \log(4/\delta)} \\
&= 2\sqrt{D_T LH \log A} + C \cdot \tau H^2 P_T + 2C_\beta \sqrt{2H^3 S^2 AT \cdot \log(dT/\delta) \cdot \log(K+1)} \\
&\quad + \sqrt{16H^2 T \cdot \log(4/\delta)} \\
&\leq 2\sqrt{D_T LH \log A} + C \cdot \tau H^2 P_T + C' \sqrt{2H^3 S^2 AT \cdot \log^2(dT/\delta)} \qquad (17)
\end{aligned}$$

where $C, C' > 0$ are universal constants, the second step holds by the definition of $\beta$, and the third step holds by the identity $T = KH$. Analyzing Equation (17) in the same way as Equation (16) in Section C (for different regimes of $P_T$) yields the result.