[Reviews · NeurIPS 2020]

Review 1

Summary and Contributions: The paper looks at developing and analyzing model free policy optimization algorithms for MDPs with non-stationary rewards, but with stationary and unknown transition matrix. Two algorithms are proposed and their regret bounds are established for different degrees of non-stationarity in the rewards. The paper provides novel contributions and is well written. ------------ Updates after rebuttal --------------------- Thank you for your responses. I have kept my score the same but I have some suggestions for the authors: 1. Practical experiments: I strongly encourage authors to include some experiments, at least in the appendix, if possible. Additionally, having a clear discussion about practical ways to deal with limitations (knowledge of P_t and D_t, a prioiri knowledge of maximum time, etc.) 2. Even if the transition dynamics is fixed, I think the issue of re-exploration would still be required when only partial feedback is available. Having a clear discussion about this in the paper could be beneficial.

Strengths: S1. Authors provide a dynamic regret analysis, as opposed to conventional static regret, which is arguably more relevant but also more involved to analyze. S2. Regret bounds are established for different degrees of non-stationarity, under assumption that the maximum degree of non-stationarity is known a-priori. Further, for the slowest changing setting, the rates are near-optimal.

Weaknesses: W1. Algorithmic novelty seems to be marginal. The algorithm seems to be heavily based on OPPO[1], albeit with a restart mechanism. W2. Full information of reward feedback is assumed. This limits the applicability of the proposed method. W3. One of the drawbacks of the algorithm is it requires knowledge of how long the algorithm will be executed, in order to set the necessary constants. Such assumptions are reasonable for stationary setting but from a practical standpoint, in a non-stationary MDP, one would like to keep the algorithm deployed online without a priori fixed finite time. W4. Can authors comment on how tight the analysis is for the setting where non-statioanrity is fast? [1] Cai, Q., Yang, Z., Jin, C., & Wang, Z. (2019). Provably efficient exploration in policy optimization. arXiv preprint arXiv:1912.05830.

Correctness: C1. The theme of the paper should be ‘non-stationary reward/cost function’ instead of ‘non-stationary environments’. The latter naturally suggests that both dynamics and rewards are changing over time, which is not the setting that this paper looks at. C2. Line 199: It is not clear why the bonus should keep decaying over time? In a non-stationary reward setting, wouldn’t there be a need for ‘re-exploration’ as well? C3. Can authors explain the design choice of not including R in the least-squares objective, but instead directly adding it when evaluating Q? C4. I find the intermediate rate of non-stationarity interesting (slow one seems almost stationary and fast one seems intractable). However, I pretty much skimmed past lines 204-210 that talk about restart during the first read. However, it seems like this restart mechanism plays a significant role in the analysis to control dynamic regret. This seems a little unintuitive to me; why should one avoid tracking in non-stationarity and instead prefer to restart 'to stabilize'? In fact, the analysis suggests to restart after every alternate episode when non-statioanrity is tending to be large.

Clarity: The paper is well written and can be published with no major changes. Minor: - Algorithms 1 and 2 should also take \alpha as input? - Confidence level \delta used in Algorithms 1 and 2 are subsumed within \beta and are perhaps not required as explicit input? - Line 541, missing reference.

Relation to Prior Work: R1. Throughout the paper it is highlighted that the proposed method is more efficient in terms of time and space. Providing more concrete discussion about this in relation to past work can be useful.

Reproducibility: Yes

Additional Feedback:


Review 2

Summary and Contributions: This paper considers the policy optimization for non-stationary RL, in which the setting is episodic MDPs with adversarial full-information reward feedback and unknown fixed transition kernels. The underlying reward distribution can be changing, so the dynamic regret is employed as the algorithm measure. The authors propose the POWER & POWER++, which is based on the algorithm of [9,18] with periodically restart. Dynamic regret bound are shown in Theorem 1 and Theorem 2.

Strengths: The setting is episodic MDPs with adversarial full-information reward feedback and unknown fixed transition kernels, while the reward distribution can be changing. The main contribution of this paper is the proposal of a novel model-free algorithm for non-stationary RL. The main idea is simply restart the method to track the non-stationary environments. The dynamic regret analysis is solid, in particular, Lemma 3 and Lemma 4 are novel, as addressed in line 277-291 (technical contribution). Another thing that I appreciate much is that the regret bound smoothly interplotes the static and dynamic scenarios. Aka, the regret bound can reduce to \sqrt{T} when P_T is small. As far as I know, previous results cannot achieve that.

Weaknesses: (1) The paper assumes a full-information reward feedback, which can be hardly thought as a realistic assumption. Instead, it would be much appreciated to consider the bandit feedback as what [1] does. (2) Moreover, the paper assumes the access of upper bounds of P_T and D_T. This is undesired in practice. There are some recent efforts in removing such dependency [2,3]. The basic idea is to run another meta bandits algorithm for selecting the optimal parameter. The techniques, in my opinion, are very possible to remove these factors. (3) The authors claim that one of the advantages of proposed algorithm comparing to previous non-stationary RL algorithms is that the algorithm is model-free and hence is more efficient in both time and space complexity. I am not aware whether the argument holds in general. I believe authors should ellaborate more on this issue. Additionally, empirical evaluations should be carried out to support the claim. -------------------- Other comment: The idea of restarting the algorithm for fighting non-stationarity is not novel. I appreciate the discussion below Algorithm 1. I'd like to add two more papers that also use restart method for handling non-stationary bandits: [4] use restart for MAB with abrupt changes; [5] use restart for linear bandits with changing regression parameter. Some minor issues: - In appendix line 541. this is a compiling error "defined in Line ?? of " Given above concerns, I can only recommend for weak acceptance. ------------------------------------ Reference [1] Efroni, Y., Shani, L., Rosenberg, A., & Mannor, S. (2020). Optimistic Policy Optimization with Bandit Feedback. arXiv preprint arXiv:2002.08243. [2] Cheung, W. C., Simchi-Levi, D., & Zhu, R. (2020). Reinforcement Learning for Non-Stationary Markov Decision Processes: The Blessing of (More) Optimism. arXiv preprint arXiv:2006.14389. [3] Cheung, W. C., Simchi-Levi, D., & Zhu, R. (2019, April). Learning to optimize under non-stationarity. In The 22nd International Conference on Artificial Intelligence and Statistics (pp. 1079-1087). [4] Allesiardo, R., Féraud, R., & Maillard, O. A. (2017). The non-stationary stochastic multi-armed bandit problem. International Journal of Data Science and Analytics, 3(4), 267-283. [5] Zhao, P., Zhang, L., Jiang, Y., & Zhou, Z. H. (2020). A simple approach for non-stationary linear bandits. In Proceedings of the 23rd International Conference on Artificial Intelligence and Statistics, AISTATS (Vol. 2020).

Correctness: I didn't check all the details. But a high-level read of the proofs, I think the proofs are correct generally.

Clarity: Yes, the paper is well-written and well-organized.

Relation to Prior Work: Please refer to my "other comment" in previous box.

Reproducibility: No

Additional Feedback: It would be better for authors to weaken the assumptions and provide some empirical studies to support their claim on the advantage of computational efficiency. *****************Post Author Rebuttal************************ I carefully go through the other reviewers' comments, and notice that we share some similar concerns: the full-inforamtion feedback setting, only reward distribution is changing without considering transition kernel, tightness of the result (especially considering the parameter-free manner of P_T). Last but not the least, I do think the paper should conduct at least some empirical analysis to support their claim on the advantage of model-free algorithms, even this paper is most theoretical.


Review 3

Summary and Contributions: The authors propose two model-free reinforcement learning algorithms for finite, episodic MDPs with adversarial rewards (full-information case) and fixed but unknown transition kernels. Upper bounds with respect to the dynamic regret are derived for the proposed algorithms.

Strengths: The contribution is mostly theoretical. The proposed algorithm and the regret bounds, while not practical, may provide new insights for future algorithms in non-stationary environments.

Weaknesses: The algorithms do not seem practical since it requires access to quantities (or their upper bounds) that are generally not available.

Correctness: I did not check the proof but the results seem plausible.

Clarity: The paper is easy to read. The presentation is mostly clear.

Relation to Prior Work: The authors have done a decent job in relating the current work to prior relevant works.

Reproducibility: Yes

Additional Feedback: Regarding the name "POWER". I am afraid there already exists another RL algorithm, named "PoWER" (see [Kober & Peters, Policy Search for Motor Primitives in Robotics, NIPS 2008]). Obviously, the authors can decide whether to insist on using the same acronym. ===== Post-Rebuttal ===== I have read the authors' feedback, I maintain my score. Thank you.

[Author Response · NeurIPS 2020]

We thank all reviewers for their positive comments. Below we first address common concerns among the reviewers, and
then respond to questions raised by individual reviewers.

**1. Response to common concerns**

-*"Knowledge of upper bounds of $P_T$ and $D_T$"*: We remark that this type of assumptions is common and standard in
literature on dynamic regret analysis of RL algorithms; see e.g. [22, 27, 39]. And even with access to upper bounds of
$P_T$ and $D_T$, it was unclear how to achieve dynamic regret bounds for policy optimization as our paper does. We do
agree that it will be interesting to investigate the setting without these assumptions; we will pursue this direction by
using the techniques developed in [12].

- *"Full-information reward feedback"*: Such assumption is standard in literature on RL problems with non-stationary
rewards; see e.g. Even-Dar et al, "Online Markov Decision Processes" (2008). Extension of our results to the case of
bandit feedback is reasonably straightforward by augmenting our algorithms with a reward estimator similar to [18].
We will explore this direction in future work.

- *"Efficiency compared to previous algorithms"*: Previous algorithms with dynamic regret guarantees are UCRL-based
and need to solve large linear programs in each step of each episode. This makes such algorithms prohibitively expensive
in computation and memory on practical problems. On the other hand, our algorithms do not require solving linear
programs and all of their steps can be computed efficiently. We will add this discussion in our final paper.

- *"Numerical experiments"*: Our paper focuses on theoretical aspects of non-stationary RL. It is an excellent suggestion
to conduct numerical experiments to support our theoretical results. We will follow up on this.

**2. Response to individual reviewers**

**Review #2**

- *"W1, algorithmic novelty"*: In addition to the restart mechanism, our Alg 2 features OMD steps for active prediction,
which helps it achieve a better dynamic regret bound than our Alg 1; see Sec 3.2, as well as Thm 2 and the remarks
beneath it for details. To the best of our knowledge, this is the first time that OMD steps are used in RL algorithms for
tackling non-stationary environments.

- *"W2, full-information reward feedback"*: Please see our responses in the previous section.

- *"W3, fixed length of execution"*: To obtain guarantees for varying execution duration, one may augment our algorithms
with a "doubling" trick commonly used in literature. We will explore this extension in future work.

- *"W4, tightness of analysis"*: When the magnitude of non-stationarity is moderate or large and $P_T$ is on the same order
of change in rewards, the results in our Thm 1 and 2 (setting $D_T = KH^3$) match those of [6] wrt the order of $T$ under
the multi-arm bandit setting, which is a special case of our episodic RL setting.

- *"C1, non-stationary environments"*: We agree that allowing varying transitions would give a more complete picture of
non-stationary environments. On the other hand, we do believe that our setting, in spite of fixed transitions, is by itself
an interesting and practical instance of non-stationary environments, as illustrated in Sec 1 and 2.3 of our paper.

- *"C2, decaying bonus over time"*: An excellent point. The purpose of bonus is to stabilize the algorithms under unknown
transitions. Since we assume fixed transitions, there is no need for re-exploration.

- *"C3, not including reward in the LS objective"*: The two ways of including rewards are equivalent. We choose the
current way as in our paper to streamline our proofs.

- *"C4, restart mechanism"*: When the level of non-stationarity is moderate or high, restarting is necessary to ensure the
learning process is not adversely affected by the irrelevant historical reward information. Another approach that serves
the same purpose is sliding window [12, 22]. Note that the master algorithm in [12] also employs a restart mechanism.

- *"R1, more efficient"*: Please see our responses in the previous section.

**Review #3**

- (1)–(3): Please see our responses in the previous section.

- *"Other comments"*: Thanks for pointing out the additional references. We will add them in our final paper.

**Review #4**

- *"Not practical, knowledge of upper bounds"*: Please see our responses in the previous section.

We appreciate the minor issues pointed out by the reviewers, and we will fix them in our final version.

[Meta-Review · NeurIPS 2020]

Reviewers generally agreed that this paper makes a good theoretical contribution towards learning in non-stationary environments, depite some shortcomings such as generality of non-stationary MDP assumption and lack of experiments.